# Phonon Pseudoangular Momentum in α-MoO_3_

**DOI:** 10.3390/nano14070607

**Published:** 2024-03-29

**Authors:** Meiqi Li, Zhibing Li, Huanjun Chen, Weiliang Wang

**Affiliations:** 1School of Physics, Sun Yat-sen University, Guangzhou 510275, China; limq57@mail2.sysu.edu.cn; 2School of Science, Shenzhen Campus, Sun Yat-sen University, Shenzhen 518107, China; stslzb@mail.sysu.edu.cn; 3State Key Laboratory of Optoelectronic Materials and Technologies, Sun Yat-sen University, Guangzhou 510275, China; chenhj8@mail.sysu.edu.cn; 4Guangdong Province Key Laboratory of Display Material and Technology, Sun Yat-sen University, Guangzhou 510275, China; 5School of Electronics and Information Technology, Sun Yat-sen University, Guangzhou 510275, China; 6Center for Neutron Science and Technology, Sun Yat-sen University, Guangzhou 510275, China

**Keywords:** phonon, pseudoangular momentum, selection rule, helicity-selective Raman scattering

## Abstract

In recent studies, it has been discovered that phonons can carry angular momentum, leading to a series of investigations into systems with three-fold rotation symmetry. However, for systems with two-fold screw rotational symmetry, such as α-MoO_3_, there has been no relevant discussion. In this paper, we investigated the pseudoangular momentum of phonons in crystals with two-fold screw rotational symmetry. Taking α-MoO_3_ as an example, we explain the selection rules in circularly polarized Raman experiments resulting from pseudoangular momentum conservation, providing important guidance for experiments. This study of pseudoangular momentum in α-MoO_3_ opens up a new degree of freedom for its potential applications, expanding into new application domains.

## 1. Introduction

In recent years, pseudoangular momentum (PAM) as a novel degree of freedom for phonons has gained widespread attention. PAM is proposed as a quantized physical quantity defined by the eigenvalue of a rotation operator [1]. Phonon PAM exhibits selective coupling with other particles/quasiparticles, such as circularly polarized light [2,3], magnetization [4], phonon Berry curvature [5,6], and chiral structures [7], all of which are expected to lead to new physical effects. Therefore, in recent years, phonon PAM has aroused great interest. Both theory and experiment have revealed their significant contributions to various optical and excitonic effects in two-dimensional (2D) semiconductors [8,9], magneto-optical responses in topological semimetals [10], valley phonon Hall effects [11], etc.

PAM has been systematically studied in systems with three-fold rotational symmetry. Zhang et al. [1] identified phonons carrying angular momentum in systems with spin–phonon interactions by applying an external magnetic field, which breaks time-reversal symmetry. Subsequently, in non-magnetic hexagonal lattice systems without an applied magnetic field, Zhang et al. [2] defined PAM to characterize angular momentum which is associated with the three-fold rotational symmetry. Related research has also been conducted in systems with four-fold rotational symmetry [12]. Zhang et al. [13] explored chiral phonons and PAM in nonsymmorphic systems, extending their work to systems with three-fold helical rotational symmetry, where they reported that PAM could be non-integer and q-dependent. In first-order Raman scattering, recent studies have revealed robust selection rules for circularly polarized Raman experiments on 2D transition metal dichalcogenides (TMDs) with three-fold rotational symmetry [3,14]. Zhang et al. [2] and Kyosuke Ishito et al. [15] found that the PAM in systems with three-fold rotational symmetry, along with the angular momentum (AM) of photons, satisfies AM conservation in circularly polarized Raman scattering.

However, for systems with two-fold screw rotational symmetry (SRS), there has not yet been any relevant discussion. α-MoO_3_ is a system with two-fold SRS. We take α-MoO_3_ as an example because of its in-plane anisotropy, making it a natural low-loss hyperbolic phonon polariton material [16,17]. It exhibits remarkable properties such as high spatial locality, slow group velocity [16], and tunability [18] of light, allowing for the compression and focusing of electromagnetic fields to very small scales [19,20]. Therefore, it has promising applications in integrated photonics chips. Study of its phonon PAM could open up a new tunable degree of freedom for its applications.

In 2021, Ali et al. carried out circularly polarized Raman experiments on α-MoO_3_ [21], but their analysis of the crystal orientation was wrong. Therefore, they failed to establish a connection between the selection rules in circularly polarized Raman experiments and the phonon PAM. Our study reveals that, for α-MoO_3_, there are no phonon modes with circular polarization at the Γ point. Therefore, to comprehend the selection rules in circularly polarized Raman spectroscopy, further investigation into the PAM of α-MoO_3_ is necessary.

In this work, we systematically examine the PAM of the system with two-fold SRS by group theory. Via revisiting the results of circularly polarized Raman experiments, we further elucidate the selection rules in circularly polarized Raman experiments of α-MoO_3_. We find that it does not correlate with the expected chiral phonon behavior (AM of phonon or so-called phonon circular polarization) but instead complies with conservation rules between the phonon PAM and photon AM. The investigation of PAM in α-MoO_3_ provides a new degree of freedom for its related applications, opening a new application domain.

## 2. Methods

We performed first-principles calculations to study the phonon properties of α-MoO_3_. The structure optimization, force constant calculations, and dielectric function calculations were conducted using density functional theory (DFT) implemented in the Vienna Ab initio Simulation Package (VASP) [22,23]. The electron core interactions were assessed using the projector-augmented wave approximation [24]. The density function was estimated using the generalized gradient approximation with the Perdew−Burke−Ernzerhof (PBE) [25] exchange-correlation potential. The energy cutoff was set to 520. Because conventional DFT does not properly account for the van der Waals (vdW) interaction, the lattice parameter perpendicular to the basal plane would be overestimated and a correction scheme is required to improve the results. Hence, optB88 was applied, which is a non-local van der Waals functional proposed by Berland and Cooper [26]. Harmonic approximation calculations of phonon frequencies and vibration vectors were obtained using the finite difference method implemented in Phonopy (2.11.0) and VASP (6.3.1). The method of nonanalytical term correction (NAC) [27] was applied.

## 3. Results

### 3.1. PAM of Phonon in Lattices with Two-Fold Screw Rotational Symmetry

Displacement of the k-th atom in the l-th unit cell can be written as:(1)$$u_{\alpha }\left(\begin{array}{c} l\\ k \end{array}\right)=\frac{1}{\sqrt{m_{k}N}}\sum_{j,\rho }\sum_{q}e^{iq\cdot R_{l}}e_{\alpha }\left(k\left|\begin{array}{c} q\\ j_{\rho } \end{array}\right.\right)Q\left(\begin{array}{c} q\\ j_{\rho } \end{array}\right)$$
where $Q\left(\begin{array}{c} q\\ j_{\rho } \end{array}\right)=Q_{0}\left(\begin{array}{c} q\\ j_{\rho } \end{array}\right)e^{\pm i\omega \left(q\left|j\right.\right)t}$ is the amplitude of the *j*-th phonon mode, ρ is the index of degeneration,$e_{\alpha }\left(k\left|\begin{array}{c} q\\ j_{\rho } \end{array}\right.\right)$ is the element of the eigenvector of the dynamical matrix, α denotes the Cartesian index,***q*** is wave vector, and ***R****_l_* is the coordinates of atoms in real space.

Each phonon mode belongs to one of the irreducible representations (IR) of the space group at ***q***. The little group determines the symmetry of the harmonic potential term of phonons, so the symmetry of the eigenfunction must be described by the little group. A little group is a group composed of all operations within the crystallographic space group that remain reciprocal lattice invariant. A symmetry operation can be written as $\hat{P}=\left\{\hat{R}|\hat{t}\right\}$, where $\hat{R}$ and $\hat{t}$ are the rotation and translation parts, respectively. The translation vector ***t*** can be further described as t=T+τ, where τ is a vector which is smaller than any primitive translation vector of the crystal, and ***T*** is a translation vector of the crystal.

In symmorphic space groups with two-fold rotation symmetry, the PAM was associated with the eigenvalue which is obtained by the ${\hat{C}}_{2}$ acting on the phonon mode.. Explicitly,
(2)$$\left\{{\hat{C}}_{2\alpha }\left|0\right.\right\}u\left(\begin{array}{c} l\\ k \end{array}\right)=exp\left[i\pi l_{\alpha }^{ph}\right]u\left(\begin{array}{c} l\\ k \end{array}\right)$$
where α is the direction of the rotation axis. The phase correlation of the phonon wave function contains two parts: one is from the local (intracell) part $e\left(\begin{array}{c} q\\ j \end{array}\right)$, and another is from the nonlocal (intercell) part $e^{iq\cdot R_{l}}$. Therefore, one can extract spin PAM $l_{\alpha }^{s}$ for the local part and orbital PAM $l_{\alpha }^{o}$ for the nonlocal part through rotation [2].

When considering a system with SRS, the PAM of phonons should be defined as follows [13]:(3)$${\hat{P}}_{\left\{R|t\right\}}u\left(\begin{array}{c} l\\ k \end{array}\right)=exp\left[-i\pi \left(l_{o}+l_{s}\right)\right]u\left(\begin{array}{c} l\\ k \end{array}\right)$$

For a system with SRS, the non-integer translation operations cannot be ignored. Let us first focus on the spin PAM $l_{\alpha }^{s}$. According to group theory analysis, the effect of a space group operation $\hat{P}=\left\{\hat{R}|\hat{t}\right\}$ applied to the eigenvector of the dynamical matrix is [28]
(4)$${\hat{P}}_{\left\{\hat{R}|\hat{t}\right\}}e_{\alpha }\left(k\left|\begin{array}{c} q\\ j_{\rho } \end{array}\right.\right)=e_{\alpha }\left(k^{\prime }\left|\begin{array}{c} q\\ j_{\rho } \end{array}\right.\right)=\sum_{k}Re_{\alpha }\left(k\left|\begin{array}{c} q\\ j_{\rho } \end{array}\right.\right)e^{iq\cdot \left(P^{-1}R_{0k^{\prime }}-R_{0k^{\prime }}\right)}\delta_{Pk,k^{\prime }}$$
where $\delta_{Pk,k^{\prime }}$ means that the atom *k* is transformed to atom *k′* by $\hat{P}$. For a little group operation ${\hat{P}}^{q}=\left\{{\hat{R}}_{q}|{\hat{t}}_{q}\right\}$, that satisfies $q=R_{q}q+G$, the element of IR is calculated by [29]
(5)$$F_{j_{\rho }{j^{\prime }}_{\rho }}\left(P_{q}\right)=\sum_{kk^{\prime },\alpha ,\beta }e_{\alpha }^{*}\left(k^{\prime }\left|\begin{array}{c} q\\ j_{\rho } \end{array}\right.\right)R_{q}^{\alpha \beta }e_{\beta }\left(k\left|\begin{array}{c} q\\ j\prime_{\rho } \end{array}\right.\right)\times e^{iq\cdot \left({P_{q}}^{-1}R_{0k^{\prime }}-R_{0k^{\prime }}\right)}\delta_{P_{q}k,k\prime }$$
where $R_{q}^{\alpha \beta }$ represents the matrix elements of $R_{q}$. When the phonon mode is *J*-fold degenerate, IR is a *J*×*J* matrix. The subscript $j_{\rho }$ is the index for the rows and columns of the IR matrix. When *J* > 1, it is worth mentioning that the IR $F\left(P_{q}\right)$ calculated by the first principle is usually not diagonalized. Thus, the eigenvector needs to be transformed to diagonalize $F\left(P_{q}\right)$ before calculating *l^s^*. After diagonalization, the $j_{\rho }$-th diagonal element $F_{j_{\rho }}^{\prime }\left(P_{q}\right)$ is the eigenvalue of the $j_{\rho }$-th degenerate phonon mode of the symmetry operations of ${\hat{P}}^{q}=\left\{{\hat{R}}_{q}|{\hat{t}}_{q}\right\}$.

The spin PAM $l_{\alpha }^{s}$ describes the rotation motion along the α-axis. Therefore, $l_{\alpha }^{s}$ is only related to atomic vibrations in the plane perpendicular to α direction. Thus, when the non-integer translation operation τ in P has a non-zero component only in the direction of α (e.g., $\hat{P}=\left\{{\hat{C}}_{2y}|\left(\begin{array}{ccc} 0 & \frac{1}{2} & 0 \end{array}\right)\right\}$), the components of $e\left(\begin{array}{c} q\\ j \end{array}\right)$ that are perpendicular to α direction can return to the original eigenstate (with an additional phase factor) after the rotational operation acting on it. Thus, the eigenvalue corresponding to the rotation operation can be obtained in this situation. Hence, only in this situation, $l_{\alpha }^{s}$ can still be defined [15]:(6)$$\left\{{\hat{C}}_{2\alpha }|0\right\}e_{\beta \gamma }\left(\begin{array}{c} q\\ j \end{array}\right)=e^{i\pi l_{\alpha }^{s}}e_{\beta \gamma }\left(\begin{array}{c} q\\ j \end{array}\right)$$

As $F_{j_{\rho }}^{\prime }\left(P_{q}\right)$ is an eigenvalue of the entire ${\hat{P}}_{\left\{R|\tau \right\}}$, we can obtain $l_{\alpha }^{s}$ after eliminating the phase factor of the non-integer translation operation $\hat{\tau }$. We first need to obtain the eigenvalues of $\hat{\tau }$ acting on the eigenvectors. Assuming ${\hat{\tau }}^{n}=\hat{T}$, where $\hat{T}$ is an integer translation term, therefore [13]:(7)$$\hat{\tau }e\left(\begin{array}{c} q\\ j \end{array}\right)={\hat{T}}^{\frac{1}{n}}e\left(\begin{array}{c} q\\ j \end{array}\right)=e^{iq\cdot \frac{T}{n}}e\left(\begin{array}{c} q\\ j \end{array}\right)=e^{iq\cdot \tau }e\left(\begin{array}{c} q\\ j \end{array}\right)$$

Hence, we obtained the eigenvalues of $\hat{\tau }$ acting on $e\left(\begin{array}{c} q\\ j \end{array}\right)$. After eliminating the phase factor of the non-integer translation operation $\hat{\tau }$ in the eigenvalue of ${\hat{P}}_{\left\{\hat{R}|\hat{\tau }\right\}}$, we can obtain the spin PAM $l_{\alpha }^{s}$ of $j_{\rho }$-th phonon mode by
(8)$$l_{\alpha }^{s}=-\frac{1}{i\pi }\left(ln\left(F_{j_{\rho }j_{\rho }}^{\prime }\left(P_{q}\right)\right)-iq\cdot \tau \right)$$

In addition, $\hat{P}$ is applied to $e^{iq\cdot R_{l}}$ as follows [13]:(9)$${\hat{P}}_{\left\{{\hat{C}}_{2}|\hat{\tau }\right\}}e^{iq\cdot R_{l}}=e^{iq\cdot \left(P^{-1}R_{l}\right)}=e^{iq\cdot \left(\left\{C_{2}^{-1}|-C_{2}^{-1}\tau \right\}R_{l}\right)}=e^{i\left\{C_{2}\right\}q\cdot \left(R_{l}-\tau \right)-iq\cdot R_{l}}e^{iq\cdot R_{l}}$$

Therefore, we can obtain the orbital PAM $l_{\alpha }^{o}$
(10)$$l_{\alpha }^{o}=-\frac{1}{\pi }\left\{\left(q-C_{2}q\right)\cdot R_{0k^{\prime }}-C_{2}q\cdot \tau \right\}$$

According to Equation (3), the total PAM is
(11)$$l_{\alpha }^{ph}=l_{\alpha }^{o}+l_{\alpha }^{s}$$

It is worth mentioning that this definition is only possible for nonsymmorphic systems with SRS whose non-integer translation operation $\hat{\tau }$ in $\hat{P}$ has a non-zero component only in the direction of α.

### 3.2. PAM of Phonon in α-MoO_3_

Based on this model, we calculate $l_{o}$ and $l_{s}$ for α-MoO_3_. α-MoO_3_ is a typical material with a two-fold SRS, belonging to the orthorhombic crystal system with space group Pnma (No.62). The crystal structure is illustrated in Figure 1a, in which crystallographic axes are shown, and Figure 1b shows the first Brillouin zone.

At Γ, it is easy to calculate the PAM of α-MoO_3_. We know $l_{\alpha }^{ph}=l_{\alpha }^{s}$ since $l_{\alpha }^{o}=0$ when q=0, according to Equation (10). At the same time, F is a scalar since the phonon modes are non-degenerate at Γ point. Generally, lattice vibrational modes can be classified based on the IR of the space group. At the Γ point, there are four IRs with Raman activity, namely $A_{g}$, $B_{1g}$, $B_{2g},$ and $B_{3g}$. The PAM values of phonon modes belonging to these four IRs at the Γ point, obtained with Equation (8), are listed in Table 1.

Beyond Γ, the PAM of α-MoO_3_ depends on wave vector ***q***. For space group Pnma, there are three symmetry operations associated with rotation, and their matrix presentations are as follows:(12)$$\left\{C_{2x}|\tau_{x}\right\}=\left(\begin{array}{cccc} 1 & 0 & 0 & \frac{1}{2}\\ 0 & -1 & 0 & \frac{1}{2}\\ 0 & 0 & -1 & \frac{1}{2} \end{array}\right),$$
(13)$$\left\{C_{2y}|\tau_{y}\right\}=\left(\begin{array}{cccc} -1 & 0 & 0 & 0\\ 0 & 1 & 0 & \frac{1}{2}\\ 0 & 0 & -1 & 0 \end{array}\right)$$
(14)$$\left\{C_{2z}|\tau_{z}\right\}=\left(\begin{array}{cccc} -1 & 0 & 0 & \frac{1}{2}\\ 0 & -1 & 0 & 0\\ 0 & 0 & 1 & \frac{1}{2} \end{array}\right)$$

The 3 × 3 matrix on the left corresponds to the rotation operation $\hat{C}$, and the last column corresponds to the non-integer translation operation $\hat{\tau }$. The three elements in the last column represent translations along the X/Y/Z direction, respectively. We can observe that the non-integer translation terms are non-zero only in the direction of the rotation axis, only in Equation (13). Therefore, for Pnma, we can define $l_{\alpha }^{s}$ only at those *q*-points whose corresponding little group possesses the $\left\{{\hat{C}}_{y}|{\hat{\tau }}_{y}\right\}$ symmetry operation. In the first Brillouin zone, this condition is satisfied only for *q*-points at four high-symmetry paths parallel to Γ−Y(DT), i.e., the B, D, P, and DT paths in Figure 1b. With group theory analysis, we can infer the characteristics of the phonon modes. On the DT path, the phonon mode degeneracy is 1, hence the IR is a scalar. The result of Equation (5) indicates that the IR $F\left(\left\{{\hat{C}}_{y}|{\hat{\tau }}_{y}\right\}\right)$ has only two values, which are $e^{i\pi q_{y}}$ and $e^{i\pi \left(q_{y}+1\right)}$, respectively. These also can be looked up from the IR table [31]. On the B path (Figure 1b), the phonon mode degeneracy is 2, thus the IR $F\left(\left\{{\hat{C}}_{y}|{\hat{\tau }}_{y}\right\}\right)$ is a 2 × 2 matrix. The result of Equation (5) and the IR table [31] show that there is only one possible IR. In this case, the IR $F\left(\left\{{\hat{C}}_{y}|{\hat{\tau }}_{y}\right\}\right)$ is $\left(\begin{array}{cc} 0 & e^{i\pi q_{y}}\\ e^{i\pi q_{y}} & 0 \end{array}\right)$, whose eigenvalues are $e^{i\pi q_{y}}$ and $e^{i\pi \left(q_{y}+1\right)}$. The results on the P and D paths are the same as those on the DT and B paths, respectively. To sum up, according to Equation (8), $l_{y}^{s}$ takes values of 0 and −1. According to Equation (10), $l_{y}^{o}$ is equal to $q_{y}$. Afterward, according to Equation (11), $l_{y}^{ph}$ takes values of $q_{y}$ and $q_{y}-1$, which is dependent on q. The values of phonon PAM along the DT high-symmetry path are shown in Figure 2.

### 3.3. Helicity-Selective Raman Scattering

Raman scattering is a powerful tool for the structural identification and characterization of materials. Circularly polarized Raman spectroscopy, by measuring the frequency shift and rotation direction of photons interacting with molecular/crystal vibrations, reveals the symmetry, configuration, and rotational properties of molecules/crystals. This technique plays a significant role in fields such as materials science, biochemistry, and pharmaceutical research, aiding scientists in understanding the behavior and properties of molecules/crystals in the microscopic world.

In the work of Shahzad Akhtar Ali et al. [21], eight distinct Raman modes were observed in helicity-selective Raman scattering from flakes of α-MoO_3_. These modes include 5 $A_{g}$ modes at 336, 364, 482, 817, and 992 cm⁻¹. The mode at 283 cm⁻¹ has the $B_{2g}$ IR, while the mode at 666 cm⁻¹ has the $B_{3g}$ IR. These designations are based on polarization-resolved Raman scattering and helical selection rules for Raman scattering. In their helicity-selective Raman scattering experiment, the flakes are oriented so that the crystallographic c- and a-axes align in-plane, and the incident light travels along the b-direction, which is perpendicular to the plane. In that study, the Raman tensors used were adapted from Ref. [32], in which the x-direction is the b-direction. However, Shahzad Akhtar Ali et al. erroneously used the y-direction as the direction of the incident wave vector in their theoretical analysis. As a result, deviations occurred in the theoretical analysis of the experimental results. Additionally, the authors failed to correlate the phonon’s PAM with the experimental results of circularly polarized Raman scattering. Therefore, in our work, we reanalyzed the experimental results and provided a theoretical explanation of the intrinsic mechanism of this physical process.

The lattice parameters of α-MoO_3_ are *a* = 13.85 Å, *b* = 3.71 Å, and *c* = 3.92 Å, with each primitive unit cell containing four Mo atoms and twelve O atoms. In this work, the layered structure of α-MoO_3_ was located in the b-c plane, designated as the y-z plane, as shown in Figure 1a.

From the phonon dispersion of α-MoO_3_ in Figure 3, we can see that there are no imaginary frequencies in the phonon dispersion. α-MoO_3_ has forty-five optical phonon branches and three acoustic phonon branches. Group theory analysis reveals that the Raman-active modes are: 8 $A_{g}$, 4 $B_{1g}$, 8 $B_{2g}$, and 4 $B_{3g}$. According to group theory analysis, the Raman tensors of $A_{g}$, $B_{1g}$, $B_{2g}$, $B_{3g}$ should have the following form [32]
(15)$$R\left(A_{g}\right)=\left(\begin{array}{ccc} a & 0 & 0\\ 0 & b & 0\\ 0 & 0 & c \end{array}\right)$$
(16)$$R\left(B_{1g}\right)=\left(\begin{array}{ccc} 0 & d & 0\\ d & 0 & 0\\ 0 & 0 & 0 \end{array}\right)$$
(17)$$R\left(B_{2g}\right)=\left(\begin{array}{ccc} 0 & 0 & e\\ 0 & 0 & 0\\ e & 0 & 0 \end{array}\right)$$
(18)$$R\left(B_{3g}\right)=\left(\begin{array}{ccc} 0 & 0 & 0\\ 0 & 0 & f\\ 0 & f & 0 \end{array}\right)$$

It is worth noting that these Raman tensors apply to the XYZ convention illustrated in Figure 1. The Raman scattering cross-section of a specific mode is proportional to ${\left|e_{s}^{*}\cdot R\cdot e_{i}\right|}^{2}$ [33,34,35,36,37,38], where $e_{i}$ and $e_{s}$ denote the polarization direction of the incident laser and the scattered light. First-principles calculations provide phonon eigenvectors at specific frequencies, which can then be used to accurately determine the IR of the phonons at those frequencies (the first column in Table 2). Typically, there is some deviation between the frequencies obtained from first-principles calculations and those measured in experiments, but this deviation remains within a reasonable range and is not excessively large. Therefore, when analyzing experimental results, it is common to initially roughly match the frequency of phonon obtained from experimental measurements with the data from first-principles calculations. Subsequently, through the analysis of experimental data from linearly polarized Raman spectroscopy and the results of Raman tensor calculations, the IR of phonon with a specific frequency is determined. Hence, errors in crystal orientation not only affected the results of circularly polarized Raman analysis but also impacted the determination of phonon IR. In the linearly polarized Raman experiments, the intensity varied as the sample rotated around the *x*-axis, which is the direction of the incident light wave vector. The intensity profiles were calculated as follows [32]:(19)$$I_{A_{g}}\left(\theta \right)\propto {\left|{\left(\begin{array}{c} 0\\ 1\\ 0 \end{array}\right)}^{T}{\left(\begin{array}{ccc} 1 & 0 & 0\\ 0 & cos\theta & -sin\theta \\ 0 & sin\theta & cos\theta \end{array}\right)}^{T}R\left(A_{g}\right)\left(\begin{array}{ccc} 1 & 0 & 0\\ 0 & cos\theta & -sin\theta \\ 0 & sin\theta & cos\theta \end{array}\right)\left(\begin{array}{c} 0\\ 1\\ 0 \end{array}\right)\right|}^{2}\;={\left|bcos^{2}\theta +csin^{2}\theta \right|}^{2}$$
(20)$$I_{B_{3g}}\left(\theta \right)\propto {\left|fsin2\theta \right|}^{2}$$
(21)$$I_{B_{2g}}\left(\theta \right)=I_{B_{1g}}\left(\theta \right)=0$$
where ${\left(\begin{array}{ccc} 0 & 1 & 0 \end{array}\right)}^{T}$ represents linearly polarized light with polarization direction along the y-axis. Our computational results and the data detected in the experiment [21] are listed in Table 2. Only phonon modes with Raman activity and frequencies between 200 cm⁻¹ and 1000 cm⁻¹ are listed. The phonon IR obtained through first-principles calculations is listed in the first column. The frequencies obtained from first-principles calculations and from experiments are listed in the second and fourth columns, respectively. The formulae in parentheses in the fourth column represent the phonon IR as proposed in work [21]. The Raman scattering intensity calculated via Equations (19)–(21) for the linearly polarized Raman experiment with light incident along the x-direction is listed in the third column.

First and foremost, it is crucial to note that Shahzad Akhtar Ali et al. [21] assert that the IR of the phonon mode with frequency of 283 cm⁻¹ is $B_{2g}$. Nevertheless, based on our reanalysis of the linearly polarized Raman experiment results on α-MoO_3_ and first-principles calculation results in Table 2, we are inclined to assert that it should be $B_{3g}$. Equations (19)–(21) suggest that only the Raman intensity of the $A_{g}$ and $B_{3g}$ modes can be observable. This implies that the phonon mode with a frequency of 283 cm⁻¹ is by no means, as they analyzed, attributed to $B_{2g}$. Through Table 2, we know that there is a phonon mode with an IR of $B_{3g}$ whose frequency is close to the mentioned frequency. Therefore, we believe that the phonon mode with a frequency of 283 $cm^{-1}$ has an IR of $B_{3g}$. In summary, in the circularly polarized Raman experiment, the authors should have detected Raman signals for three $A_{g}$ modes (at 336, 817, and 992 cm⁻¹) and two $B_{3g}$ modes (at 283 and 666 cm⁻¹).

Furthermore, we analyzed the experimental results of circularly polarized Raman spectroscopy. Firstly, we calculated the Raman scattering intensity of the circularly polarized. Circularly polarized light is divided into left-handed and right-handed categories, commonly described using helicity and also indicated by spin AM. When the wave vector direction is along the *x*-axis, left-circularly polarized light is represented as ${\left(\begin{array}{ccc} 0 & 1 & i \end{array}\right)}^{T}$, with spin AM $m_{photon}=-1$, and right-circularly polarized light is represented as ${\left(\begin{array}{ccc} 0 & 1 & -i \end{array}\right)}^{T}$, with spin AM $m_{photon}=+1$. Similar conventions apply to the other two directions. Our computational results are listed in Table 3. The column groups with the background colors yellow/green/red in Table 3 correspond to the light’s AM along the X/Y/Z axes, respectively, with the propagation direction of both incident and scattered light also along the X/Y/Z axes. The underlines indicate modes with strong Raman intensity were detected in the experiment [21]. The experimental results are consistent with our prediction of the Raman intensity.

The behavior of circularly polarized light in Raman scattering can be predicted by the Raman tensor [33]. However, the classical theory cannot tell us the intrinsic mechanism. Thus, it is an important issue to study the PAM of phonons and the conservation law of the PAM of phonons [2,8] in a crystal.

In existing circularly polarized Raman experiments, samples are probed using circularly polarized light, then the circular polarization of the scattered light is detected. For some modes of phonon, the helicity of the scattered light is opposite to that of the incident light, with $△m_{photon}=\pm 2$ (✔️ in the second column in each colored group in Table 3). For some other modes of phonon, the helicity of the scattered light is the same as that of the incident light, with $△m_{photon}=0$ (✔️ in the first column in each colored group in Table 3). Since the incident light and the scattered light travelled along the x and the -x direction, respectively, it can be seen that only the Raman scattering intensity of the $A_{g}$ and $B_{3g}$ modes could be non-zero, because the PAM of these phonon modes along x direction is 0 ($l_{x}^{ph}=0$); the Raman scattering intensity of the B_1g_ and B_2g_ modes should be zero, because the PAM of these phonon modes along x direction is 1 ($l_{x}^{ph}=1$). A similar analysis holds for the other two directions. Based on the computed Raman intensity (Table 3) and experimental results, we can infer that as a selection rule for the optical transition process, PAM offers an additional conservation condition besides the crystal momentum conservation and energy conservation. Due to momentum matching requirements, we generally consider phonons at the Γ point to couple with light in terms of AM, which follows the selection rule [39] in a crystal with two-fold SRS,
(22)$$m_{i}-m_{s}=-l_{\alpha }^{ph}+2p\left(p=0,\pm 1\right)$$
where $m_{s}$ and $m_{i}$ represent the AM of the scattered and incident photons, respectively. Thus, the selection rule can be described as follows: if the sum of photon AM and phonon PAM is not conserved, the Raman effect cannot occur. If the sum of photon AM and phonon PAM change is conserved, the Raman effect may occur.

## 4. Conclusions

In our work, the PAM in a crystal with two-fold SRS was systematically examined. For α-MoO_3_, PAM is only identifiable along the high-symmetry path parallel to Γ−Y. The PAM values on the Γ−Y path are associated with *q*-points, taking values of $q_{y}$ and $q_{y}-1$. Furthermore, due to momentum matching requirements, only the phonons at the Γ point can be coupled with light. The PAM values at the Γ point were examined and the selection rules in circularly polarized Raman experiments of α-MoO_3_ were clarified through theoretical analysis and examination of experimental results. The selection rules in circularly polarized Raman experiments of α-MoO_3_ do not conform to previous expectations related to chiral phonons but rather comply with conservation rules on the sum of phonon PAM and photon AM. For circularly polarized incident light and scattered light, $△m_{photon}\;$ = 2 or $△m_{photon}\;$ = 0, only phonons with PAM = 0 have non-zero Raman scattering intensity, while phonons with PAM = 1 always have zero Raman scattering intensity. For α-MoO_3_, when the AM of the incident light is along the out-of-plane direction, only $A_{g}$ and $B_{3g}$ modes exhibit non-zero Raman scattering intensity. This investigation of PAM in α-MoO_3_ paves the way for its potential applications.

## Figures and Tables

**Figure 1 nanomaterials-14-00607-f001:**
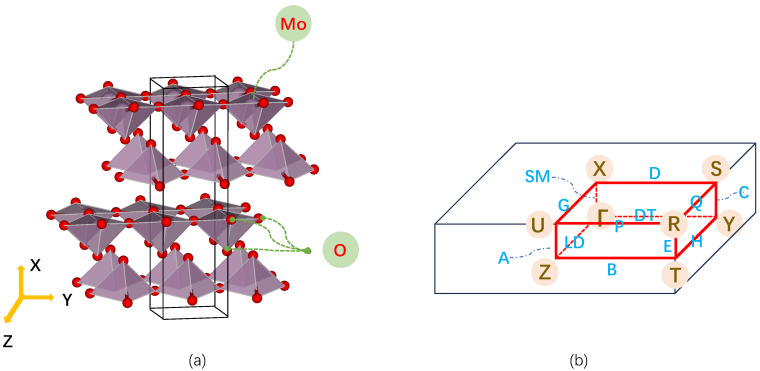
(**a**) Atomic structure of α-MoO_3_. The red balls represent oxygen atoms, forming the vertices of the pyramid, with a molybdenum atom at the center of the pyramid. The black rectangular prism indicate the unit cell. (**b**) First Brillouin zone of α-MoO_3_ [30]. The blue text indicates the lables of high-symmetry point paths, while the brown text within circles indicates the lables of high-symmetry points. The first Brillouin zone is represented by a red rectangular prism.

**Figure 2 nanomaterials-14-00607-f002:**
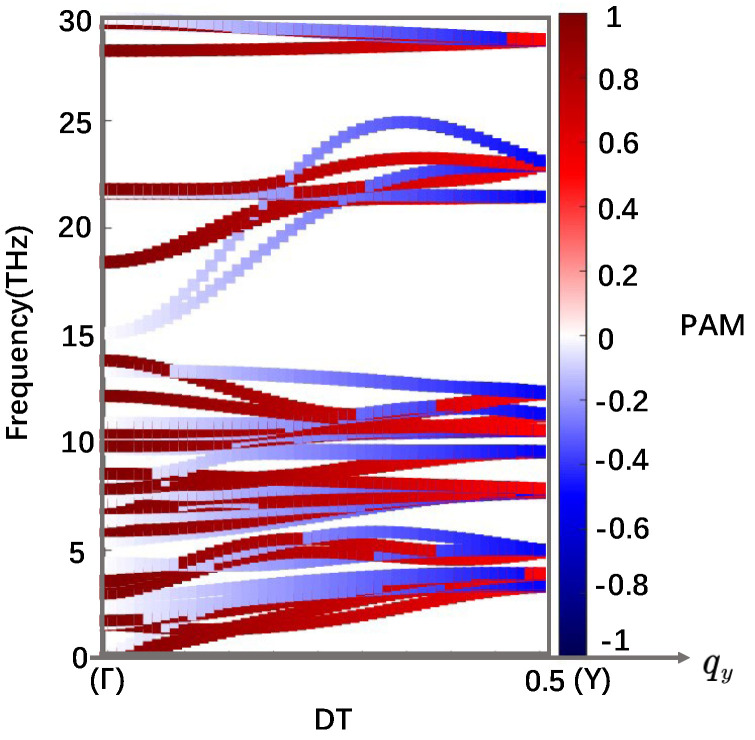
PAM of phonons along the Γ−Y (DT) high-symmetry path, $\left(q_{x}=q_{z}=\;0\right)$. The color axis represents PAM.

**Figure 3 nanomaterials-14-00607-f003:**
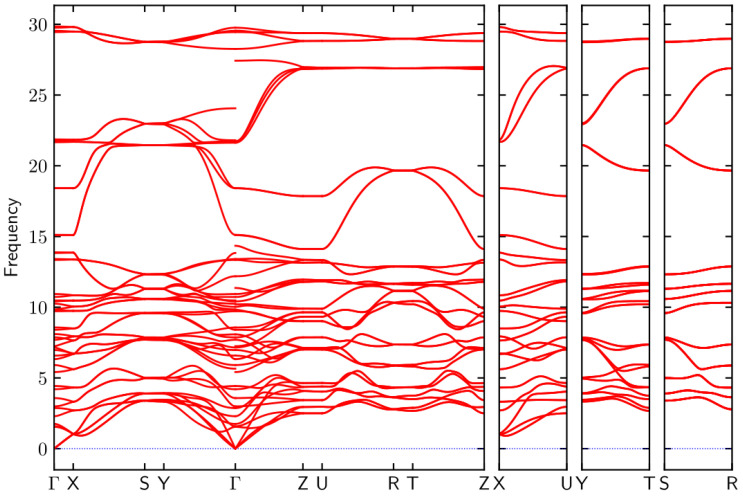
Phonon dispersion of α-MoO_3_.

**Table 1 nanomaterials-14-00607-t001:** PAM of α-MoO_3_ at Γ point.

	$l_{x}^{ph}$	$l_{y}^{ph}$	$l_{z}^{ph}$
*A_g_*	0	0	0
*B* _1*g*_	1	1	0
*B* _2*g*_	1	0	1
*B* _3*g*_	0	1	1

**Table 2 nanomaterials-14-00607-t002:** Comparison of experimental and calculated phonon frequencies (in cm⁻¹) and IR. ✔️ represents that the Raman intensity can be non-zero, and ❌ indicates that the Raman intensity should be zero.

IR	Frequency from First-Principles Calculations	Raman Intensity	Frequency from Experiment [21]
$A_{g}$	219	✔️	
$B_{1g}$	239	❌	
$B_{2g}$	257	❌	
$B_{3g}$	262	✔️	283 ($B_{2g}$)
$B_{1g}$	286	❌	
$A_{g}$	324	✔️	336 ($A_{g}$)
$B_{2g}$	328	❌	
$A_{g}$	347	✔️	364 ($A_{g}$)
$B_{2g}$	365	❌	
$A_{g}$	447	✔️	482 ($A_{g}$)
$B_{2g}$	461	❌	
$B_{3g}$	614	✔️	666 ($B_{3g}$)
$B_{1g}$	614	❌	
$A_{g}$	725	✔️	817 ($A_{g}$)
$B_{2g}$	727	❌	
$A_{g}$	985	✔️	992 ($A_{g}$)
$B_{2g}$	996	❌	

**Table 3 nanomaterials-14-00607-t003:** Raman intensity depending on the electric polarization. ✔️ represents that the Raman intensity can be non-zero, and ❌ indicates that the Raman intensity should be zero.

Incident Laser	$\left(\begin{array}{ccc} 0 & 1 & i \end{array}\right)$		$\left(\begin{array}{ccc} 1 & 0 & i \end{array}\right)$		$\left(\begin{array}{ccc} 1 & i & 0 \end{array}\right)$	
Scattered light	$\left(\begin{array}{c} 0\\ 1\\ i \end{array}\right)$	$\left(\begin{array}{c} 0\\ 1\\ -i \end{array}\right)$	$\left(\begin{array}{c} 1\\ 0\\ i \end{array}\right)$	$\left(\begin{array}{c} 1\\ 0\\ -i \end{array}\right)$	$\left(\begin{array}{c} 1\\ i\\ 0 \end{array}\right)$	$\left(\begin{array}{c} 1\\ -i\\ 0 \end{array}\right)$
*A_g_*	✔️	✔️	✔️	✔️	✔️	✔️
*B* _1*g*_	❌	❌	❌	❌	✔️	✔️
*B* _2*g*_	❌	❌	✔️	✔️	❌	❌
*B* _3*g*_	❌	✔️	❌	❌	❌	❌

## Data Availability

The original contributions presented in this study are included in the article; further inquiries can be directed to the corresponding authors.

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
