# Peer review of "Phonon Pseudoangular Momentum in α-MoO3"

_nanomaterials, 2024, doi:10.3390/nano14070607_

Round 1
Reviewer 1 Report
Comments and Suggestions for Authors
The study of chiral phonons or phonon pseudoangular momentum (PAM) has been intensively studied only recently and Raman spectroscopy is one of the basic experimental tool in this field.
In this manuscript, the authors theoretically study PAM in a-MoO2 with 2-fold screw rotational symmetry. The selection rules for Raman scattering on phonons measured with linearly polarized radiation can nowadays be standardly determined (and the authors present them here as well), but the authors also determined the selection rules in circularly polarized Raman experiment, which is not common in the literature and is a major contribution of this work. Authors also compare the mode activities with previously published Raman spectra and explain the discrepancy with the results in ref. 31 due to the erroneous crystal orientation of MoO3 in ref. 31.
The paper is written clearly, lucidly and I did not find any large inaccuracy in it. Maybe only three small remarks:
1. Page 2, raw 71: What is cutoff 520? Please, explain it.
2. Page 2, raw 89, there should be probably “little group” instead of “litter group”.
3. Ref. 14 needs revision.
I recommend the paper for publication after minor changes mentioned above.
Reviewer 2 Report
Comments and Suggestions for Authors
The article is devoted to the study of pseudoangular momentum in α-MoO3. The research is aimed at an interesting and important topic. The presentation of the material is consistent. However, the text contains points that need to be reworked before publication.
Lines 210-216 discuss the irreducible representations of the phonon mode with frequency of 283 cm⁻¹ in comparison with work [19]. The claim that it is B3 and not B2 is not explained properly. Moreover, the comparison of the calculation carried out by the authors with the experiment carried out in work [19] is discussed on page 8 very concisely. The authors should describe in more detail the experimental data themselves, with which they compare their calculation, and explain the intropretation of experimental data made in work [19] and the intropretation of experimental data made by the authors.
The introduction should provide more detailed information about the subject of the study and about similar works, if such are published.
Reviewer 3 Report
Comments and Suggestions for Authors
The authors presented original study on phonon pseudoangular momentum in α-MoO3. By investigating the unique characteristics of α-MoO3, researchers have uncovered a novel aspect of phonon behavior in materials with 2-fold screw rotational symmetry. Through a systematic discussion grounded in group theory, the study elucidates the significance of pseudoangular momentum conservation in circularly polarized Raman scattering. The findings not only contribute to fundamental knowledge in materials science but also pave the way for potential applications in integrated photonics chips. This research underscores the importance of considering crystal symmetry in the design and analysis of optical and excitonic effects in nanomaterials. The exploration of phonon pseudoangular momentum opens up new avenues for manipulating light at small scales, offering promising prospects for advanced photonics technologies. The study's rigorous approach and experimental validation enhance our understanding of the intricate quantum phenomena governing optical responses in materials with specific rotational symmetries. Overall, this research represents a significant step forward in the field of nanomaterials, showcasing the power of combining theoretical insights with experimental observations to unravel complex physical phenomena.
Several comments can be made on the article:
1. There is no testing of the obtained theoretical expressions on experimental results. It is necessary to add information about comparing the results obtained with known data.
2. The conclusion must be rewritten according to existing standards. Authors often use the verb discuss. It is not suitable for describing the results of work
3. English needs to be improved
After these minor changes, the article can be published.
Comments on the Quality of English Language
1. The conclusion must be rewritten according to existing standards. Authors often use the verb discuss. It is not suitable for describing the results of work
2. English needs to be improved
Reviewer 4 Report
Comments and Suggestions for Authors
See the pdf file attached.

Round 2
Reviewer 2 Report
Comments and Suggestions for Authors
In the revised version of the article, the authors responded to the reviewer's comments and made valuable changes to the text. The article can be published in its present form.